# SAFE: Improving LLM Systems using Sentence-Level In-generation Attribution

## Abstract

Large Language Models (LLMs) are increasingly applied in various science domains, yet their broader adoption remains constrained by a critical challenge: the lack of trustworthy, verifiable outputs. Current LLMs often generate answers without reliable source attribution, or worse, with incorrect attributions, posing a barrier to their use in scientific and high-stakes settings, where traceability and accountability are paramount. To be reliable, attribution systems require high accuracy for short-length attribution on retrieved data, i.e., attribution to a sentence within a document rather than the entire document. We propose SAFE, a Sentence-level Attribution FramEwork for Retrieve-Augmented Generation (RAG) systems that attributes generated sentences during generation. This allows users to verify sentences as they read them and correct the model when the attribution indicates the generated text is not grounded in the documents, increasing the safety of LLM systems. This framework consists of two steps: predicting the required number of references for a sentence, and attributing the sentence. Our approach achieved 95% accuracy in the first step, which translated to 2.1∼6.0% improvements in the accuracy (normalized for maximum possible accuracy) of all attribution algorithms in our clean dataset, when compared to their top-1 accuracy. We also applied SAFE in real-world scenarios with documents containing hundreds to thousands of sentences. In these settings, SAFE reliably attributed sentences to their source documents, demonstrating that the method generalizes beyond controlled benchmarks. The SAFE framework and the training dataset are publicly available on GitHub[1].

## 1 Introduction

Recent advances in the field of machine learning enabled the creation of deep learning models that are not limited to specific tasks but are general-purpose, capable of performing a wide range of functions at near-human performance levels. Large Language Models (LLMs) (Brown et al., 2020) are widely accessible and often provide accurate answers to general queries. However, they are also prone to generating incorrect statements (Gravel et al., 2023; Li et al., 2024a; Zhang et al., 2025; Bang et al., 2025), which can be risky in systems where accuracy is essential. Despite their shortcomings, LLMs are driving progress across science and industry. Yet, users in domains where correctness and accuracy are of high importance, e.g. medical, law, and academic scholarly, are hindered by the possibility of the generated content being incorrect. Improving LLMs' trustworthiness and verifiability becomes a critical challenge, especially when it is difficult to judge the correctness of the answer (Sadeghi et al., 2024).

Attribution enhances the trustworthiness of LLMs by identifying the sources that influence their outputs (Rashkin et al., 2022; Yue et al., 2023), improving transparency and enabling fact-checking (Chen et al., 2024). We detect two key issues with LLM-generated responses in the context of retrieval-augmented generation (RAG) systems: (1) answers that contain false statements, and (2) references that are missing, unrelated, or hallucinated (Zuccon et al., 2023). Research shows that users are more likely to trust systems that provide references, even if they are incorrect, compared to those that provide no references at all (Bansal et al., 2021; Eiband et al., 2019; Lai and Tan, 2019). This highlights the risk of users placing undue trust in attribution systems that generate inaccurate references, reinforcing the need for reliable attribution. Moreover, popular LLM interfaces often cite

---

[1]https://anonymous.4open.science/r/SAFE-ICLR

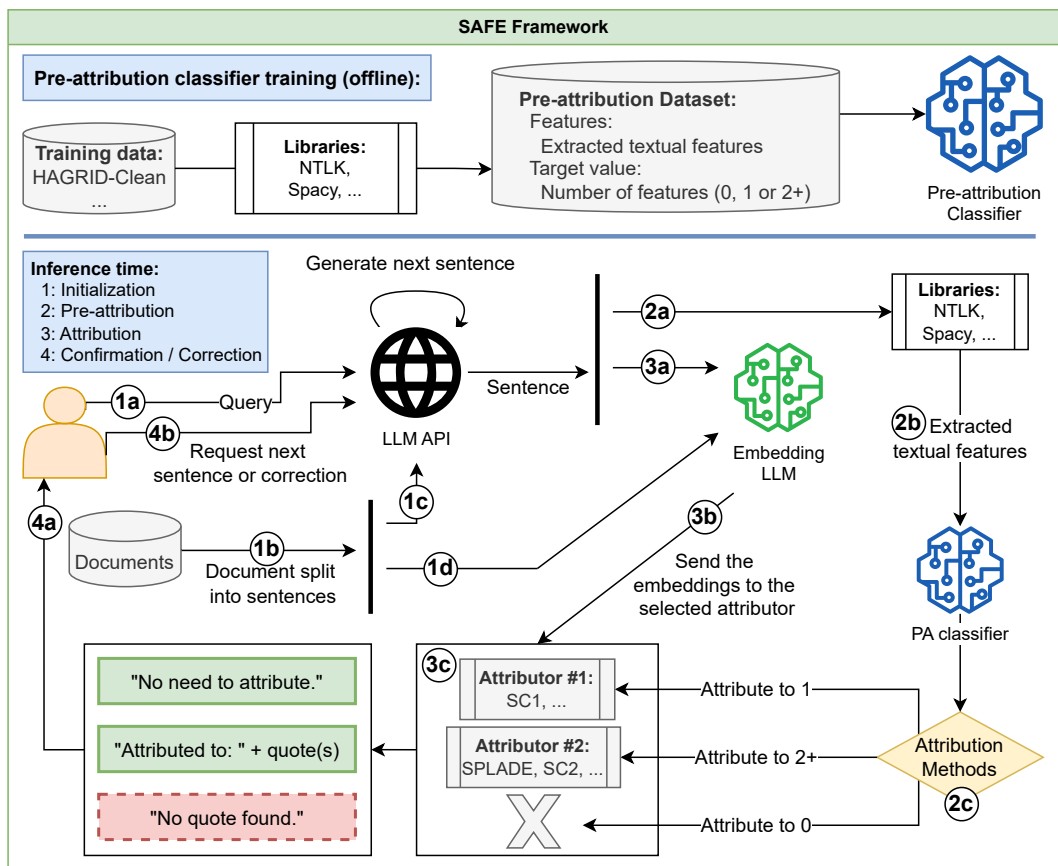

Figure 1: SAFE Framework. Before running SAFE, a pre-attribution classifier is induced to predict if a sentence should be attributed to 0, 1, or multiple quotes. During inference, the user provides a query and documents to the system; a LLM API is used to generate an answer, sentence by sentence; the PA classifier assigns the sentence to an attributor; and the attributor returns a quote (if possible and necessary) to the user; lastly, the user may ask for the next sentence or correct the LLM.

entire documents as sources (e.g., Perplexity (2023)), requiring users to read the full text to verify claims. This makes fact-checking in such systems impractical (Min et al., 2023; Yue et al., 2023).

The lack of trust in LLMs results in agencies such as the 117th United States Congress (2021-2022), the European Commission (2021), and the Japanese AI Safety Institute (AISI) limiting their use in high-risk applications, leading to scrutiny to improve the reliability of LLMs.

To address current attribution limitations, we provide a clean version of the HAGRID attribution dataset Kamalloo et al. (2023), and we propose SAFE, a sentence-level in-generation attribution framework for RAG systems (see Figure 1). In its current implementation, this framework allows the user to quickly verify if a generated sentence is true by showing a sentence extracted from the RAGged document that has the same meaning. Failure to provide an attributable sentence indicates that the generated text may not be grounded in the document. Since this implementation is attributing during text generation, the user may correct the LLM if the text being generated does not match the document, allowing for corrections before the whole text is generated. We apply SAFE to real-world scenarios where the search space has hundreds to thousands of sentences, and the results suggest that the method generalizes beyond the control benchmarks. The logs from these chats are seen in the appendix section of this document.

**Using SAFE:** While a more detailed description is given in the Section E (appendix); to use SAFE, the user only needs to follow these steps: download the repository; run *setup_HAGRID_Clean.py*; open *SAFE_FullSystem/*; add the RAG documents to *documents/*; add their API key to *Conversational_GPT.py* and run *main.py*. Booting SAFE may take a while on the first boot, depending on the associated downloads. We intend to make SAFE available through *pip install* soon.

## 2 BACKGROUND AND RELATED WORK

This work focuses on sentence-level attribution in Retrieval Augmented Generation (RAG) (Lewis et al., 2020) systems. As such, the background is connected to the research fields of attribution in LLMs and sentence-level Information Retrieval (IR). Additionally, while we do not propose a reasoning model, in-generation attribution can contribute to faithful reasoning (Creswell and Shanahan, 2022; Tan et al., 2024) by verifying that each reasoning step is grounded on fact.

### 2.1 ATTRIBUTION IN LLMS

Attribution can be broadly split into two categories based on when the source information is accessed: at training or inference time. While training-time attribution has been explored (Koh and Liang, 2017; Grosse et al., 2023; Chang et al., 2024), this work is focused only on inference-time attribution, i.e., RAG. The increasing need for reliable attribution has made "Attribution in LLMs" an increasingly popular research area, with several methods surveyed by Li et al. (2023). Following this survey, we broadly categorize attribution into three categories:

**Model-driven attribution:** The model uses its internal knowledge to answer questions and provide sources. Since the sources are generated, they are often hallucinations themselves (Agrawal et al., 2024), making this approach not reliable;

**RAG:** Using RAG (Lewis et al., 2020; Schick et al., 2023; Asai et al., 2023; Bašaragin et al., 2024; Radhakrishnan et al., 2024), before generating an answer, the algorithm uses the input query to retrieve information from an external source and integrates it into the query. With this step, the answer will be based on both the model's internal knowledge and the retrieved information. While theoretically speaking, the answer can be attributed back to the RAGged document, the internal knowledge of the LLM may conflict with the external knowledge, resulting in a hallucinated answer (Xie et al., 2024). As a consequence, the answer does not match the referenced documents;

**Post-generation Attribution:** The answer is generated using the model's internal knowledge or using RAG. After generation, it uses an information retrieval algorithm to fetch external references that back the generated answer (Nakano et al., 2022; Qin et al., 2023). SAFE fits into this category since, although it uses a RAG approach to generate text, attribution is done after generating each sentence, verifying whether or not its generation was based on the document.

### 2.2 SENTENCE-LEVEL INFORMATION RETRIEVAL

On a high level, the objective of this work is to, given a sentence and a document, obtain a quote from the document that matches the sentence. This task is related to the field of information retrieval. By providing direct quotations from the source documents, we can guarantee that the explanation is as true as the source document, reducing the time spent on fact-checking from reading a whole document to a single sentence. Sentence retrieval has been studied for several decades (Aho and Corasick, 1975; Manning et al., 2008; Karpukhin et al., 2020), including methods from the most naive (e.g., exact or fuzzy string matching) to modern approaches that employ LLMs for semantic similarity using embeddings, keyword search, or dense passage retrieval:

**String Matching:** Both exact and fuzzy string matching (Navarro, 2001) aim to find an exact or similar matching of the input sentence in the document. Although simple and computationally efficient, it is not reliable to use this approach for attribution;

**Semantic Similarity using Embeddings:** Calculating the embeddings of a sentence using an LLM (Devlin et al., 2018; Liu et al., 2019; Sanh et al., 2019; Wang et al., 2020), reduces the sentence to a point in an $n$-dimensional space. In this space, sentences with similar meanings are placed closer to each other, allowing the association of sentences that are similar in meaning, even when the sentences are written in different ways. While this approach allows us to capture semantic meaning, calculating embeddings is computationally expensive;

**Dense Passage Retrieval (DPR):** Similarly to semantic similarity with embeddings, DPR (Karpukhin et al., 2020) splits documents into sections and ranks them based on similarity with the input. The main difference is the size of the sections retrieved, which, due to their larger size, require that the user read a longer excerpt of text to validate the generated sentence;

**Keyword Search (KS):** KS (Manning et al., 2008; Formal et al., 2021) is a middle point in the amount of information extracted and the computational cost. By extracting keywords from sentences, two sentences are considered similar if the same set of keywords is extracted from them. These keywords can be words present in the sentences or synonyms.

We extend previous work on information retrieval on string matching, semantic similarity, and keyword search by applying it to attribution in LLMs, providing a sentence-level attribution system that verifies the truthfulness of a generated sentence by searching for equivalent sentences in documents. It should be noted that, since SAFE performs attribution in-generation, we require fast attribution algorithms to minimize the waiting time of the user. This motivates the "pre-attribution" step, later described, which enables SAFE to select an attribution algorithm based on the generated sentence.

# 3 METHODOLOGY

This work introduces SAFE, a sentence-level in-generation attribution framework for RAG systems that, given a query and documents, generates the output sentence by sentence, while associating each generated sentence with a sentence present in the input documents, whenever necessary and possible. By providing quotes from the documents to the user during generation, the user can easily verify the generated content and ask for clarifications to the LLM if the answer is something unexpected. All the code used in this paper is available at GitHub (hidden-for-anonymity reasons). As seen in Figure 1, this system divides attribution into two steps: selecting the attributor algorithm and using it. We refer to the first step as *"pre-attribution"*. This step aims to optimize the quality of the attribution pipeline by first identifying how many references should be assigned to each sentence. In this sense, a sentence may be simple enough for a classifier to consider that (a) it does not require attribution at all, (b) it should be attributed to a single sentence, or (c) it should be attributed to multiple sentences.

The remainder of this section describes the datasets used to train and validate the SAFE framework, how we preprocessed the dataset, what classification and attribution algorithms were used, and also how we apply it to real-world scenarios.

## 3.1 WEBGLM-QA, HAGRID, AND HAGRID-CLEAN DATASETS

SAFE is initialized by inducing a classifier capable of predicting the correct number of references a sentence should have. To do this, we experiment with the WebGLM-QA, HAGRID, and HAGRID-Clean datasets. HAGRID-Clean is a version of HAGRID that was manually cleaned and labeled by us, as explained below. While other attribution datasets exist, such as TabCite (Mathur et al., 2024), ASQA (Stelmakh et al., 2022), ELI5 (Fan et al., 2019), EquinorQA (Garigliotti et al., 2024), we focus on HAGRID (Kamalloo et al., 2023) and WebGLM-QA (Liu et al., 2023), as both are ready-to-use sentence-level attribution datasets. Both datasets contain a similar structure; thus, while we primarily describe HAGRID below, this also applies to WebGLM-QA. See Table 1 for additional details.

Each entry in the dataset comprises three elements: a query, one or two answers, and a list of quotes that can be attributed to each sentence in the answers, as seen in Table 2. In this work, we are only concerned with the answers (sentences to attribute) and references; the queries are not used. Each answer comes split into a list of sentences, and each sentence is associated with a list of references (if any). Each entry contains up to 12 attributable quotes, and the sentences contain up to 9 references. The displayed sample contains two attributable quotes and two answers. The first answer comprises a single sentence, with no references. The second answer contains two sentences: the first is attributed to both quotes, and the second is attributed to only the first quote.

Table 1: Details on the WebGLM-QA, HAGRID, and HAGRID-Clean datasets

| | No. Queries | No. Quotes | No. Sentences (No. samples) | No. references in each sentence | | | Label (Ideal no. references) | | |
|---|---|---|---|---|---|---|---|---|---|
| | | | | 0 | 1 | 2+ | Zero | Single | Multiple |
| **WebGLM-QA** | 43979 | 3-5 | 186027 | 31.3k | 118.0k | 36.6k | Same as no. references | | |
| **HAGRID** | 2638 | 1-12 | 7702 | 714 | 5455 | 1533 | Same as no. references | | |
| **HAGRID-Clean** | 2638 | 1-12 | 7308 | 808 | 4838 | 1662 | 403 | 6140 | 765 |

Table 2: Example of a sample within the HAGRID dataset.

| Query | What does it mean to be an evergreen tree? |
|---|---|
| Quotes | [1] Trees are either evergreen, having foliage that persists and remains green... 
 [2] In botany, an evergreen is a plant that has leaves throughout the year, always... |
| Answer #1 
 Answer #2 | To be an evergreen tree means to have foliage that persists and remains green... 
 It is a plant that has leaves throughout the year and never completely loses... [1,2] 
 Most conifers, including pine and fir trees, are evergreens, while deciduous... [1] |

Table 3: Sentences within HAGRID. The classifier used for pre-attribution makes a prediction that does not match the labelled number of references, but we consider the prediction correct.

| Type of Issue | Sentence |
|---|---|
| Over referencing | Scotland's national dish is haggis [1][2][4]. |
| Under referencing | The atomic number of mercury is 80. |
| Under referencing | However, it was later acquired by Google in September 2009. Luis von Ahn was the originator of the reCAPTCHA program [2] |
| Invalid sentence | Kearney/Foreign Policy Magazine Globalization Index.[1] |

### 3.1.1 PROCESSING OF THE HAGRID DATASET

During our preliminary experiments, we encountered several challenges with both the HAGRID and WebGLM-QA datasets that affected model accuracy. These included noisy input data and inconsistencies between the content of sentences and the number of listed references. While WebGLM-QA is too large for manual cleaning, we manually curated HAGRID to produce a cleaned version, which we refer to as HAGRID-Clean.

**Cleaning Samples:** Cleaning the samples was a straightforward process that involved standardizing the reference format and merging duplicated sentences. As seen in Table 2, the sentences in the datasets include the reference list. We went through all samples, separating each sentence into a tuple containing the sentence itself and the list of references (now written consistently among all samples). Many samples contained duplicated sentences, i.e., sentences are shown multiple times, with different references. Merging them reduced the number of sentences within the dataset from 7702 to 7308.

**Cleaning Labels:** We found that the raw number of references per sentence is a misleading way to determine how many references a sentence should have, as illustrated in Table 3. Some sentences were over-referenced. Despite being simple, they included unnecessarily long lists of references, which can mislead models into interpreting them as information-rich. We also observed under-referenced sentences, where complex or factual statements were not supported by enough references. Lastly, some sentences were invalid, containing no attributable content.

To address these issues, we reviewed each sentence in HAGRID and assigned a new label indicating whether it should have zero, one, or multiple references or be marked as invalid. For training purposes, invalid sentences were grouped with the "zero" class to increase class diversity. In addition to label noise, we observed inconsistent referencing styles across samples[2], which made it non-trivial to extract citations accurately. These inconsistencies affected both HAGRID and WebGLM-QA.

### 3.2 PREPROCESSING THE DATASETS FOR PRE-ATTRIBUTION AND ATTRIBUTION

### 3.2.1 INPUT VARIABLES

**Pre-attribution:** As shown in Figure 1, the first step in our pipeline involves splitting the HAGRID and WebGLM-QA datasets into individual sentences to extract input variables for the classification model. In this work, we create two different sets of feature, making two versions of each dataset. In the first version, for each sentence, we compute 24 numerical features that capture various textual properties, such as sentence length, reading ease, and the number of named entities, among others.

---

[2]Referencing styles found: [1], [1][2], [1,2], [1, 2], [1,2,], [1 and 2], [1-2], (1), and [context 1], among others.

The full list of features can be seen in Table 10, in the appendix. We hypothesize that we can use these features to measure the information complexity of a sentence with enough accuracy to predict the number of references it should contain.

In the second version of the dataset, we use the embeddings of each sentence as input features. Using this approach, we use the *all-MiniLM-L6-v2* to convert sentences to their embedding, an array with 384 dimensions. As such, in this version, the pre-attribution datasets have 384 features.

**Attribution:** The pre-processing approach used for attribution is highly dependent on the attribution algorithm used, so further details will be provided in the attribution algorithms section below.

### 3.2.2 TARGET VALUES

**Pre-attribution:** The goal of the pre-attribution step is to classify each sentence into one of three categories, based on the number of references it ideally requires: zero, one, or multiple. In the original HAGRID and WebGLM-QA datasets, this labeling process was automatic. We assigned labels based on the number of citations already linked to each sentence. For the HAGRID-Clean dataset, as previously mentioned, we introduced an additional layer of supervision by assigning labels based on human judgment. Instead of relying solely on citation count, we assessed each sentence's informational content to determine how many references it should have, correcting for over- or under-referencing in the raw data.

### 3.2.3 TARGET VALUES FOR ATTRIBUTION AND EVALUATION METRIC

Each sentence $i$ in is accompanied by three elements: a list of references used to generate the answer $\mathcal{R}_i$, the target list of references to which the sentence can be attributed to $T_i$, and a label indicating the number of references the sentence should be attributed to $L_i$ Taking this information as input, the attribution algorithm predicts a set of references to attribute the sentence to, $P_i$.

For sentence $i = 1, \ldots, N$ let:

$$T_i \subseteq \mathcal{R}_i, \quad t_i = |T_i|, \qquad P_i \subseteq \mathcal{R}_i, \quad p_i = |P_i|, \qquad L_i \in \{\text{ZERO}, \text{ONE}, \text{MULTIPLE}\}.$$

Taking this into consideration, the correctness indicator used for attribution is defined as:

$$\text{Corr}(i) = \begin{cases} 1, & \text{if } L_i = \text{ZERO} \wedge P_i = \emptyset, \\ 1, & \text{if } L_i = \text{ONE} \wedge \big((p_i = 1 \wedge P_i \subseteq T_i) \vee (t_i = 0 \wedge P_i = \emptyset)\big), \\ 1, & \text{if } L_i = \text{MULTIPLE} \wedge \big((p_i \geq 2 \wedge P_i \subseteq T_i) \vee (t_i < 2 \wedge P_i = T_i)\big), \\ 0, & \text{otherwise.} \end{cases}$$

Finally, the evaluation metric is simply:

$$\text{Accuracy} = \frac{1}{N} \sum_{i=1}^{N} \text{Corr}(i).$$

### 3.3 CLASSIFICATION ALGORITHMS FOR PRE-ATTRIBUTION

We experimented with four classifiers to predict how many references a sentence should have: Random Forest (RF), XGBoost (XGB), Multi-Layer Perceptron (MLP), and TabularNet (TN). We used the implementations provided by the *sklearn* and *xgboost* Python libraries. Since the dataset is highly unbalanced, we changed the $class\_weight$ parameter of all classifiers to $balanced$. The full list of parameters is displayed in the appendix, in Table 6. We evaluate the classifiers through 30 independent runs per classifier-dataset pair. Each run uses a different training and test split, with 70% of the samples allocated for training and 30% for testing, while maintaining the original class ratio.

### 3.4 ATTRIBUTION ALGORITHMS

We use six attribution algorithms: two embedding-based attribution algorithms implemented by us to be used as a baseline, fuzzy string matching, BM25, SPLADE, and MMR. It should be noted that, except for Fuzzy and SPLADE, all of the following methods always return at least one quote, making

Table 4: Training and test accuracy (%) (30-run average) in each test case in pre-attribution.

|  | HAGRID | HAGRID-Clean | WebGLM-QA |
|---|---|---|---|
| **Textual Features** | | | |
| Random Forest | 99.5 / 71.5 | 99.1 / 95.6 | 99.2 / 63.9 |
| XGBoost | 99.7 / 67.5 | 99.7 / 95.0 | 99.8 / 60.4 |
| MLP | 80.3 / 66.7 | 97.1 / 94.7 | 65.2 / 65.2 |
| TabNet | 95.0 / 63.6 | 98.0 / 94.6 | 66.2 / 65.2 |
| **Embeddings** | | | |
| Random Forest | 99.8 / 71.2 | 100.0 / 84.5 | 99.8 / 63.7 |
| XGBoost | 99.8 / 72.2 | 100.0 / 90.8 | 96.1 / 56.9 |
| MLP | 99.8 / 64.0 | 100.0 / 86.1 | 73.3 / 60.0 |
| TabNet | 99.8 / 63.9 | 100.0 / 82.7 | 74.3 / 60.8 |

them unable to detect when there are no matching quotes. This will be discussed in the results section. While we acknowledge that other methods have been developed (Gao et al., 2023; Momii et al., 2024; Berchansky et al., 2024; Li et al., 2024b; Zhao et al., 2025), we focus on these attributors due to their efficiency, which allows them to be used in-generation, with minimal latency to the user.

**Select Closest One:** This method (SC1) attributes a sentence to the single most similar quote, using the cosine distance in the embedding space as a similarity metric. This method is computationally efficient and yields good results, but cannot assign multiple references.

**Select Closest Two:** This method (SC2) works similarly, but, in addition to searching for single quotes, it also searches for pairs of quotes by calculating the distance between the sentence and the average point of the embeddings of all pairs of quotes, returning one or two quotes.

**Fuzzy String Matching:** This method uses the *rapidfuzz* Python library to calculate the fuzz ratio between the sentence and all quotes, retrieving the two most similar sentences. We include a similarity threshold to detect when no similar sentences are available in the search space.

**BM25** (Trotman et al., 2014) is a ranking function used in information retrieval that estimates how relevant a document is to a given query. This relevance is calculated based on string matching, taking into consideration if the document contains the query words and how often they appear, while giving less importance to common words.

**SPLADE** (Formal et al., 2021) represents text as a sparse vector of words, where each word is given a learned weight indicating its importance. The vector is expanded by predicting related terms, allowing the search for similar words that are not in the input sentence. SPLADE applies this procedure to all sentences in the document to search for similar sentences using the dot product as a distance metric.

**MMR** (Carbonell and Goldstein, 1998) is an embedding-based algorithm that uses cosine similarity as a distance metric. MMR promotes the diversity of the selected sentences to avoid retrieving near-duplicated sentences, making it a good candidate for an attributor that retrieves multiple sentences.

## 4 RESULTS

### 4.1 ACCURACY ON PRE-ATTRIBUTION

This section focuses on the results obtained by the RF, XGB, MLP, and TN classifiers in the two versions of each dataset. In Table 4, we see the average training and test accuracy, obtained over 30 runs in each test case. Figures 2 to 7 (appendix) show the respective confusion matrices.

Interestingly, in the HAGRID dataset, the results are consistent between using textual features and embeddings as dataset features, indicating that embeddings are a reasonable approach for this task, since they can be extracted automatically. Nevertheless, on the clean dataset, it becomes more apparent that textual features lead to better results, with all classifiers obtaining approximately the same test accuracy. Additionally, in the original dataset, models tend to classify most samples as ONE, leading to the illusion of a good model. In the Clean version of the dataset, all models show good results in all three classes. Interestingly, although MLP tends to classify most samples as

Table 5: Accuracy of each attribution algorithm when using the true label of the pre-attribution (PA) dataset to dictate the desired number of quotes (roofline); top-1 accuracy, and accuracy when using XGB for PA of each algorithm (normalized to the roofline). The XGB results in this table are a 30-run average. Table 13 (appendix) also includes the HAGRID and WebGLM-QA results.

| Attribution Method: | SC1 | SC2 | BM25 | MMR | Fuzzy | SPLADE |
|---|---|---|---|---|---|---|
| Number of sentences returned: | 1 | | 1, 2 | | 0, 1, 2 | |
| **HAGRID-Clean** | | | | | | |
| Attribution accuracy (True Label for PA) | | | | | | |
|    Accuracy on the dataset | 76.44 | 77.91 | 71.69 | 75.96 | 59.56 | 78.11 |
| Attribution top-1 accuracy | | | | | | |
|    Normalized | 92.78 | 91.03 | 91.31 | 93.36 | 90.31 | 93.79 |
| Attribution accuracy (XGB for PA) | | | | | | |
|    Test accuracy (normalized) | 97.87 | 96.98 | 95.95 | 95.67 | 94.81 | 95.88 |
|      Improvement vs top-1 | **+5.09** | **+5.96** | **+4.64** | **+2.30** | **+4.50** | **+2.09** |

ONE in the HAGRID and WebGLM-QA datasets when using textual features, it seems to be able to separate the classes to some degree when using embeddings as features.

The WebGLM-QA dataset suffers from the same problem as HAGRID: the lack of clean data causes low accuracy in test data. Additionally, since this dataset is 25 times larger than the HAGRID datasets, training a model here takes much longer, which led us to limit the MLP and TN algorithms to 2000 iterations, resulting in low training accuracy. As seen in the confusion matrices (Figure 4), despite MLP and TN obtaining the highest accuracy in this dataset, they classify most sentences as ONE.

## 4.2 ACCURACY ON ATTRIBUTION

Taking into consideration that all classifiers obtain similar results in the HAGRID-Clean dataset during attribution, and that XGB outperforms the other classifiers in terms of predictions per second (see Table 12), we focus this section on comparing three different approaches: attribution using top-1 (no pre-attribution), relying on XGB for pre-attribution, and using the labeled value. The last approach simply serves as a way to measure the maximum theoretical accuracy of the attribution algorithm by assuming a perfect classifier in pre-attribution.

Table 5 shows the results obtained using each attribution method (SC1, SC2, Fuzzy, BM25, SPLADE, and MMR) in the HAGRID-Clean dataset. We see that, despite the simplicity of this method, selecting the top-1 sentence using SC1 provides reasonable results at a low computational cost. While these results are further improved by SC2 and SPLADE, despite their popularity, Fuzzy and MMR are left behind. Besides having a borderline better accuracy, unlike the other methods, SPLADE uses a word-based vector database that allows the user to see what words are being used to match two sentences, bringing explainability to the attribution process. As additional motivation for pre-attribution, BM25's accuracy using the true label in the WebGLM-QA dataset is 96.74%, with its top-1 accuracy being 64.39% of that value (see Table 13 in the appendix).

## 4.3 REAL-WORLD APPLICATIONS

Our ultimate goal is to provide a framework that enables quick fact-checking of information generated by LLM systems. As such, we also include in this work four real-world application examples using the OPEN AI API (*gpt-4.1-mini* and *gpt-5-mini* models). While we use the OpenAI API, the framework accepts any LLM system. We test the SAFE framework for four tasks: querying about medicine consumption, summarizing scientific papers, querying scientific papers, and a small reasoning task where GPT-5-mini is prompted to explain and calculate Earth's gravity using two sources. We provide the full chat transcripts for these tasks in Section E (appendix).

Since we intend to attribute generated text *in-generation*, we query the model using the following template, allowing the user to verify and correct each sentence before the LLM is allowed to continue, guiding generation in the user's intended path. "DONE" is used as a keyword to stop the algorithm:

```
[QUERY] Reply one sentence at a time and wait for me to say ok before
showing the next sentence.  After answering, say DONE.
```

Below, we see a transcript of the interaction between the *user* and the *system*. While the attribution algorithm failed to correctly attribute the sentence on the first try, the user prompts the system to check the document, resulting in a new sentence being generated and correctly attributed. This shows that while the framework is not correct 100% of the time, working on a sentence level allows us to easily guide the answer into making it easier for attribution to work – or correct the model if no evidence is found.

```
I'm not feeling well, and my doctor said I should take a pill of paracetamol.
Is it safe to take another pill?  Reply one sentence at a time and wait for
me to say ok before showing the next sentence.  After answering, say DONE
ChatGPT: You should not take more than 2 tablets of paracetamol (500 - 1000
mg) at a time.
System:  Attributed to -- "You should take no more than 8 paracetamol con-
taining tablets or capsules (4000mg or 4g of paracetamol) in 24 hours ."
USER: Is that in the document?
ChatGPT: Yes, the document states, "You should not take more than 2 tablets
at a time."
System:  Attributed to -- "You should not take more than 2 tablets at a
time." [...]
```

## 5    CONCLUSIONS AND LIMITATIONS

This work introduces SAFE, a sentence-level in-generation attribution framework for RAG systems that can be used in low-end machines. Using this framework, the generative LLM answers a query, sentence by sentence, allowing us to attribute its output to the RAGged document and verify its truthfulness before moving to the next sentence. With this framework, both experts and the general public can inquire the LLMs and both verify the quality of the answer and prompt for further details as the answer is being generated. This framework has two steps to attribute a sentence: deciding how many quotes a sentence should be attributed to, and searching for that amount of matching quotes in the document. Obtaining a classifier that can correctly classify sentences into the required number of references was not trivial since the available attribution datasets have not been cleaned. To address this issue, we picked a small dataset from the literature and cleaned it, making it available at our GitHub repository, together with the SAFE framework.

Our results demonstrate that the classifiers correctly predict how many references a sentence should have 95% of the time in the cleaned dataset, raising the attributors' accuracy by 2.1∼6.0%, when compared to their top-1 accuracy. To further validate this framework, we applied SAFE in real-world scenarios with documents that far surpass the search space in the datasets, demonstrating that SAFE generalizes beyond controlled benchmarks.

**Limitations:** Sentence-level attribution has its own issues, e.g., the retrieved sentences may be selected out of context. While we did not find this problem during our experiments, we are aware that this is an issue to be addressed. Additionally, although the Fuzzy and SPLADE methods can detect if no quotes match the input sentence, they are very limited in this task, motivating the search for other attributors. The datasets used for pre-attribution suffer from class imbalance, resulting in low robustness in the pre-attribution model. We intend to explore semi-supervised learning techniques to increase the training data, compensating for this issue. Manually labeling datasets, describing how many references a sentence should have, introduces a bias. Our idea of single-reference sentences may differ from that of another person. A good example of this scenario is seen in Table 3; does "the atomic number of mercury is 80" require a reference, or should it be considered common knowledge? We took a conservative approach and decided that this and similar sentences need a reference, resulting in fewer sentences with the "ZERO" label when compared to the original dataset.

### SOFTWARE AND DATA

The HAGRID (Kamalloo et al., 2023) and WebGLM-QA (Liu et al., 2023) datasets can be accessed through their respective papers. Our code and the clean version of HAGRID are available on GitHub[1].

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

CONTENTS

## A    LLM USE IN THIS RESEARCH

LLMs were used for four tasks in this work:

**Generating Text:** We use *gpt-4.1-mini* and *gpt-5-mini* through OpenAI's API for the real-work application experiments. The first model was used to answer queries about research papers and medicine use, given a document. And the second model was used for a reasoning task to calculate the force of gravity given two sources;

**Calculating Embeddings:** We use the *all-MiniLM-L6-v2* model to calculate embedding values for our attribution algorithms and to generate the embedding-based version of the pre-attribution datasets;

**Retrieval of Related Work:** We use LLMs, through their web apps (ChatGPT and DeepSeek), to bootstrap our search for related work;

**Polishing the Document:** Lastly, we use LLMs, through their web apps (ChatGPT and DeepSeek), to help us polish the first draft of the document. After this, the document was heavily rewritten so the influence of LLMs should be vestigial.

## B    REPRODUCIBILITY STATEMENT

### B.1    ENVIRONMENT

Our experiments were conducted on machines with the specifications described below.

**Machine #1:**
**Hardware:**  Quadro RTX 8000 x 2, Intel(R) Xeon(R) Gold 5215 CPU @ 2.50GHz, 512GB RAM
**Software:**  Ubuntu 24.04.3 LTS, Python 3.12.3, PyTorch(2.7.0), CUDA(12.4), scikit-learn(1.5.2)

**Machine #2:**
**Hardware:** RTX 2000 Ada Generation Laptop GPU, Intel(R) Core(TM) i7-13800H, 32GB RAM
**Software:**  Linux Mint 22 Wilma, Python 3.12.3, PyTorch(2.7.1+cu118), CUDA(12.4), scikit-learn(1.5.2)

### B.2    IMPLEMENTATION AND PARAMETERS

The random forest, xgboost, and multilayer perceptron implementations were obtained from the *sklearn* library. The TabNet implementation was obtained from the *pytorch_tabnet* library. The SelectClosest attribution algorithms use our implementation and the *all-MiniLM-L6-v2 model*, BM25 uses the implementation in the rank_bm25 library, and the SPLADE algorithm uses the *splade-cocondenser-selfdistil* model. For the real-world application experiments, we use OpenAI's API and the *gpt-4.1-mini* model.

#### B.2.1    PRE-ATTRIBUTION

Table 6 shows the parameters used for the experiments in the pre-attribution section of this paper. Note that in the attribution section of the results, we focus exclusively on the XGBoost classifier induced using the HAGRID-Clean dataset and the textual features.

#### B.2.2    ATTRIBUTION

The attribution section of the paper presents six different attribution approaches: (1) selecting the closest embedding or (2) the closest pair of embeddings, (3) selecting the closest one or two sentences using a fuzzy approach, (4) using the BM25 algorithm, and (5) using the SPLADE algorithm. None of these approaches requires training data or has parameters, so there is no additional information to disclose.

Table 6: Parameters used for pre-attribution using textual features and sentence embeddings as dataset features. Non-disclosed parameters use their default value.

|  | Textual Features | Sentence Embeddings |
|---|---|---|
| **Random Forest** | | |
| max_depth | 10 | 16 |
| class_weight | "balanced" | "balanced" |
| **XGBoost** | | |
| max_depth | 10 | 8 |
| n_estimators | 50 | 300 |
| sample_weight | (balanced on *fit()* using the *compute_sample_weight* function) | |
| **Multilayer Perceptron** | | |
| hidden_layer_sizes | (64, 32) | (64, 32) |
| activation | "relu" | "relu" |
| solver | "lbfgs" | "lbfgs" |
| max_iter | 500 | 20000 |
| learning_rate | "constant" | "adaptive" |
| learning_rate_init | 0.001 | 0.1 |
| **TabNet** | | |
| n_d | 8 | 8 |
| n_a | 8 | 8 |
| n_steps | 2 | 2 |
| gamma | 1.5 | 1.5 |
| lambda_sparse | 1e-5 | 1e-7 |
| mask_type | "entmax" | "entmax" |
| optimizer_params | dict(lr=2e-2) | dict(lr=2e-3) |
| max_epoch | 200 | 1000 |

### B.3 RUNNING EXPERIMENTS

It should be noted that the hardware used has no impact on the accuracy values on the pre-attribution and attribution steps. It only impacts the speed at which the models are trained and the predictions.

All experiments testing the accuracy of the pipelines were run in Machine #1. Since one of our objectives is to provide a system that the general public can use, the real-world application experiments were performed in Machine #2.

## C DATASETS

### C.1 WEBGLM-QA AND HAGRID

Both of these datasets have a similar structure, exemplified in Table 7. Each sample is composed of a single query, one or two answers, and a list of quotes to which the sentences in each answer can be attributed. The answer comes pre-split into sentences, so the only preprocessing required is making a script to extract the references from the sentences and use them as labels for attribution.

### C.2 HAGRID-CLEAN

Table 8 examplifies the same sentence, now in the HAGRID-Clean dataset. The advantages of using this dataset are that now the references are in a different field from the sentence, and that now there is an additional field "type" that states the ideal number of references in a sentence. Note that HAGRID is very inconsistent with the referencing format[3]. So, having a separate and uniform field with the references is very useful. Table 9 shows a sentence that, while it has 3 references, we consider 1 reference to be enough to validate the statement.

---

[3]Referencing styles found: [1], [1][2], [1,2], [1, 2], [1,2,], [1 and 2], [1-2], (1), and [context 1], among others.

Table 7: Example of a query within the HAGRID dataset.

```
1  {"query_id": 3193,    "query": "Where was Socrates born?",
2  "quotes": [{
3          "idx": 1, "docid": "25664190#15",
4          "text": "Socrates was born in Alopeke, and belonged to the tribe
               Antiochis. His father was Sophroniscus, a sculptor, or
               stonemason. His mother was a midwife named Phaenarete.
               Socrates married Xanthippe, who is especially remembered for
               having an undesirable temperament. She bore for him three
               sons, Lamprocles, Sophroniscus and Menexenus."
5      },{
6          "idx": 2, "docid": "8564570#2",
7          "text": "Alexander Socrates Onassis was born at the Harkness
               Pavilion, a private clinic in New York City's NewYork\u2013
               Presbyterian Hospital. He was the elder child of the Greek
               shipping magnate Aristotle Onassis (1906\u00a0\u20131975) and
                his first wife, Athina Livanos (1929\u00a0\u20131974),
               herself a daughter of a Greek shipping magnate, Stavros G.
               Livanos. Onassis was named after his father's uncle, who was
               hanged by a Turkish military tribunal during their sacking of
                Smyrna in September 1922. Onassis's sister, Christina, was
               born in 1950."}],
8  "answers": [{
9          "answer": "Socrates was born in Alopeke, belonging to the tribe
               Antiochis. [1]",
10         "answer_type": "long",
11         "informative": 1,
12         "attributable": 1,
13         "sentences": [
14             {
15             "text": "Socrates was born in Alopeke, belonging to the tribe
                   Antiochis. [1]",
16             "answer_type": "long",
17             "index": 0,
18             "attributable": 1,
19             "informative": 1
20             }
21         ]
22     },{
23         "answer": "Socrates was born in Alopeke [1].",
24         "answer_type": "short",
25         "informative": 1,
26         "attributable": 1,
27         "sentences": [
28             {
29             "text": "Socrates was born in Alopeke [1].",
30             "answer_type": "short",
31             "index": 0,
32             "attributable": 1,
33             "informative": 1
34             }
35  ]}]}
```

## C.3 FEATURES FOR PRE-ATTRIBUTION

Our pre-attribution step deals with inducing and using a classifier that, given a sentence, can predict what is the ideal number of references it should have. To do this, we try two different approaches for feature engineering: using textual features, such as reading difficulty, the length of the sentence, among other features displayed in Table 10; and also a more straightforward approach based on the embedding space. The idea behind the second approach was that, since we are already calculating the

Table 8: Example of a query within the HAGRID-Clean dataset.

```
1  {
2  "query": "Where was Socrates born?",
3  "quotes": [{
4      "quote": "Socrates was born in Alopeke, and belonged to the tribe
           Antiochis. His father was Sophroniscus, a sculptor, or stonemason
           . His mother was a midwife named Phaenarete. Socrates married
           Xanthippe, who is especially remembered for having an undesirable
            temperament. She bore for him three sons, Lamprocles,
           Sophroniscus and Menexenus."
5      },{
6      "quote": "Alexander Socrates Onassis was born at the Harkness
           Pavilion, a private clinic in New York City's NewYork-
           Presbyterian Hospital. He was the elder child of the Greek
           shipping magnate Aristotle Onassis (1906-1975) and his first wife
           , Athina Livanos (1929-1974), herself a daughter of a Greek
           shipping magnate, Stavros G. Livanos. Onassis was named after his
            father's uncle, who was hanged by a Turkish military tribunal
           during their sacking of Smyrna in September 1922. Onassis's
           sister, Christina, was born in 1950."
7      }],
8  "answers": [{
9      "answer": [{
10         "sentence": "Socrates was born in Alopeke, belonging to the tribe
              Antiochis.",
11         "references":"[1]",
12         "type":"Single-reference "}]
13     },{
14     "answer": [{
15         "sentence": "Socrates was born in Alopeke. ",
16         "references":"[1]",
17         "type":"Single-reference "
18     }
19  ]}]  }
```

Table 9: Example of a query within the HAGRID-Clean dataset. While this sentence has 3 references, we consider it only requires one reference to back the statement.

```
1  [...]
2  {
3  "sentence": "The red imported fire ant, which is the most commonly
      encountered species in the United States, is originally from
      Argentina with populations found in various South American countries.
       ",
4  "references":"[4, 5, 6] ",
5  "type":"Single-reference "}]}}
6  }
```

embeddings of all sentences for the attribution step of the pipeline, why not use this information in the dataset?

Table 10: Features extracted using the *spacy*, *textstat*, *textblob*, and *nltk* Python libraries.

| ID | Description |
|----|-------------|
| 0  | Fraction of unique words (lexical diversity). |
| 1  | Named entity density: fraction of tokens that are named entities. |
| 2  | Syntactic parse tree depth. |
| 3  | Flesch reading ease score (higher = easier to read). |
| 4  | Shannon entropy of character or word distribution. |
| 5  | Average number of WordNet synsets per word (semantic ambiguity). |
| 6  | Ratio of nouns to verbs. |
| 7  | Proportion of stopwords in the sentence. |
| 8  | Ratio of punctuation marks to total words or characters. |
| 9  | Average number of characters per word. |
| 10 | Total number of syllables in the sentence. |
| 11 | Total number of words. |
| 12 | Number of unique (distinct) words. |
| 13 | Average bigram probability (lower = less expected). |
| 14 | Average trigram probability (lower = less expected). |
| 15 | Ratio of pronouns to total words. |
| 16 | Ratio of verbs in passive voice. |
| 17 | Binary indicator if the sentence is a named entity. |
| 18 | SMOG index (grade level). |
| 19 | Coleman–Liau index (readability score). |
| 20 | Automated Readability Index. |
| 21 | Dale–Chall readability score. |
| 22 | Linsear Write readability formula. |
| 23 | Gunning Fog index. |

# D  EVALUATION METRICS

## D.1  ACCURACY AND EFFICIENCY ON PRE-ATTRIBUTION

Table 11 shows the average training and test accuracy values obtained over 30 runs in each test case. The best results are seen on the HAGRID-Clean dataset, where all models have similar accuracy values in the test set. Taking into consideration the performance of each model, measured as the number of predictions per second (see Table 12), we pick XGBoost as our preferred classifier, since its speed is 1.36x faster than the best classifier while having a small difference in accuracy.

Table 11: Training and test accuracy (30-run average) of each classifier in each test case in pre-attribution. (As seen in the main text)

|                    | HAGRID      | HAGRID-Clean | WebGLM-QA   |
|--------------------|-------------|--------------|-------------|
| **Textual Features** |           |              |             |
| Random Forest      | 99.5 / 71.5 | 99.1 / 95.6  | 99.2 / 63.9 |
| XGBoost            | 99.7 / 67.5 | 99.7 / 95.0  | 99.8 / 60.4 |
| MLP                | 80.3 / 66.7 | 97.1 / 94.7  | 65.2 / 65.2 |
| TabNet             | 95.0 / 63.6 | 98.0 / 94.6  | 66.2 / 65.2 |
| **Embeddings**     |             |              |             |
| Random Forest      | 99.8 / 71.2 | 100.0 / 84.5 | 99.8 / 63.7 |
| XGBoost            | 99.8 / 72.2 | 100.0 / 90.8 | 96.1 / 56.9 |
| MLP                | 99.8 / 64.0 | 100.0 / 86.1 | 73.3 / 60.0 |
| TabNet             | 99.8 / 63.9 | 100.0 / 82.7 | 74.3 / 60.8 |

## D.2  ACCURACY ON ATTRIBUTION

Table 12: Number of thousands of predictions per second of each model in each dataset, measured on Machine#1.

|  | HAGRID | HAGRID-Clean | WebGLM-QA |
|---|---|---|---|
| **Textual Features** | | | |
| Random Forest | 19.2 | 28.3 | 38.0 |
| XGBoost | 29.8 | 38.6 | 180.4 |
| MLP | 57.2 | 20.9 | 140.1 |
| TabNet | 34.6 | 10.4 | 43.2 |
| **Embeddings** | | | |
| Random Forest | 6.5 | 7.1 | 23.7 |
| XGBoost | 21.3 | 6.5 | 43.5 |
| MLP | 19.2 | 11.9 | 41.1 |
| TabNet | 53.4 | 27.9 | 59.4 |

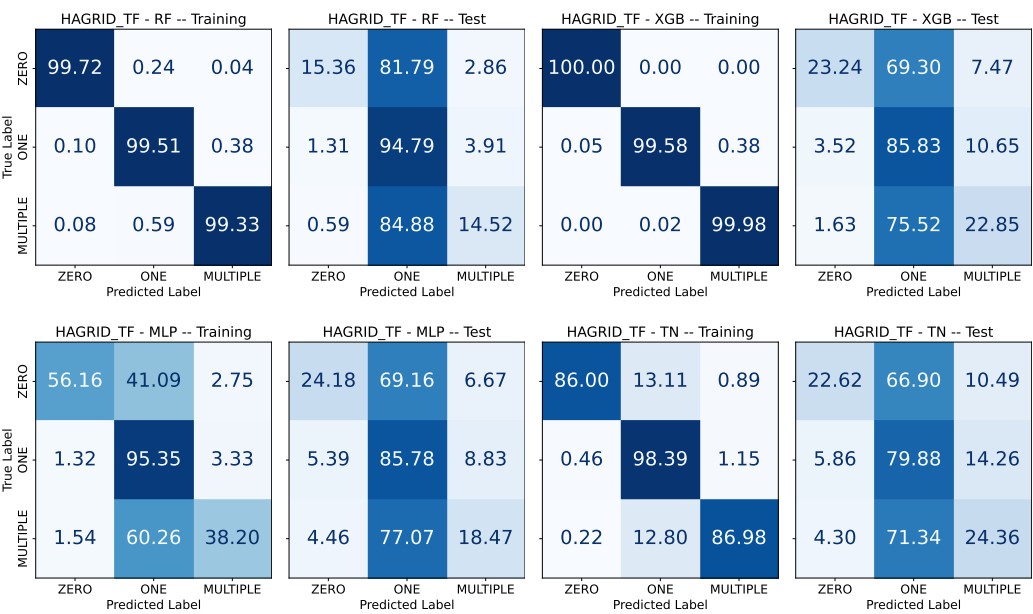

Figure 2: Average confusion matrices obtained in each experiment. The matrices have been normalizes in each row, highlighting the accuracy in each class.

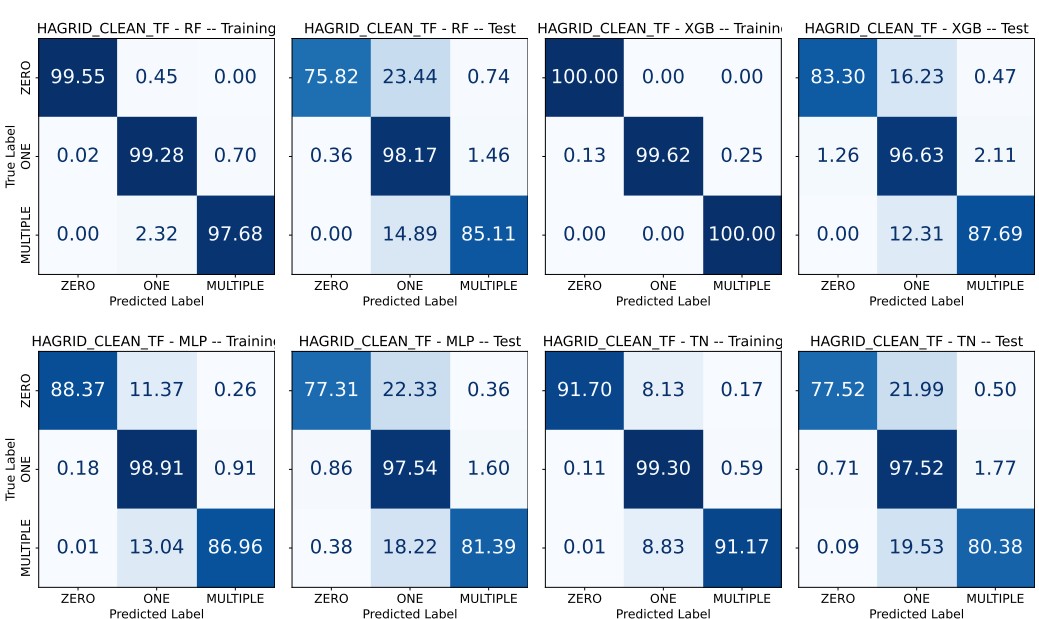

Figure 3: Average confusion matrices obtained in each experiment. The matrices have been normalizes in each row, highlighting the accuracy in each class.

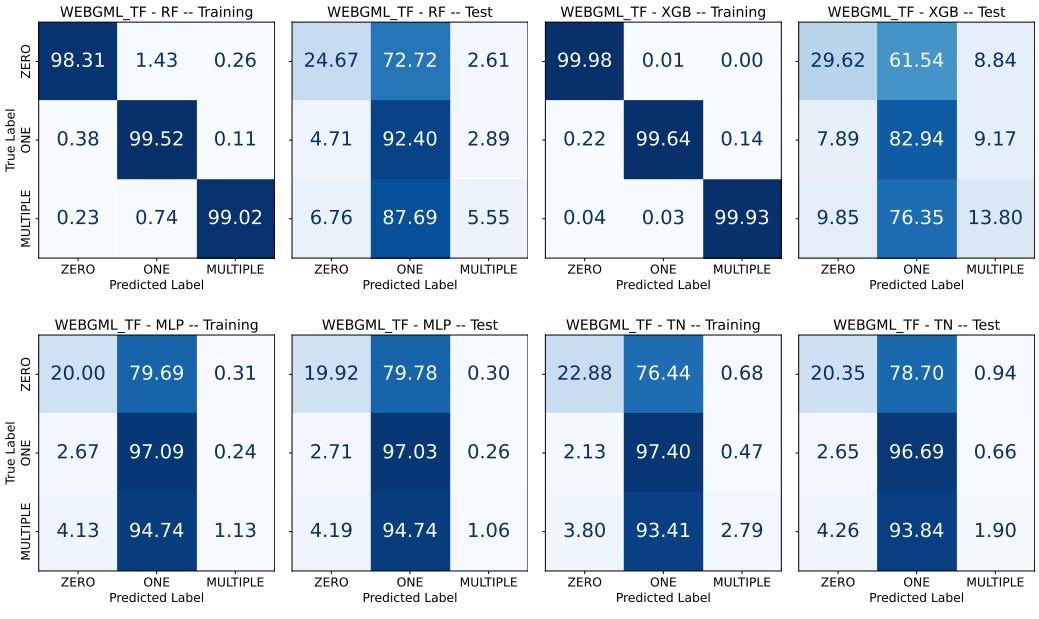

Figure 4: Average confusion matrices obtained in each experiment. The matrices have been normalizes in each row, highlighting the accuracy in each class.

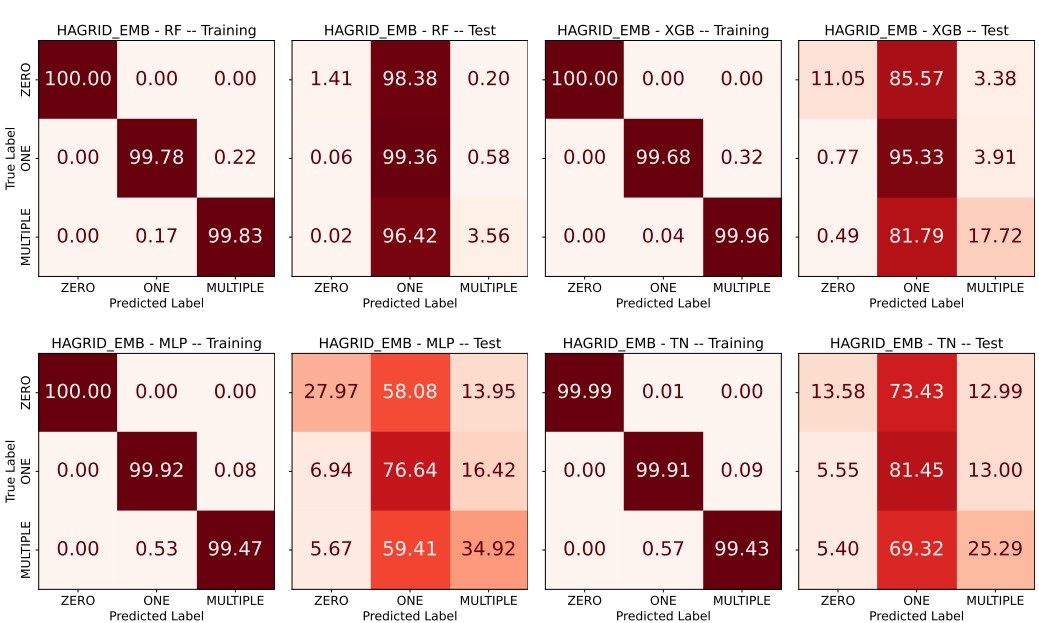

Figure 5: Average confusion matrices obtained in each experiment. The matrices have been normalizes in each row, highlighting the accuracy in each class.

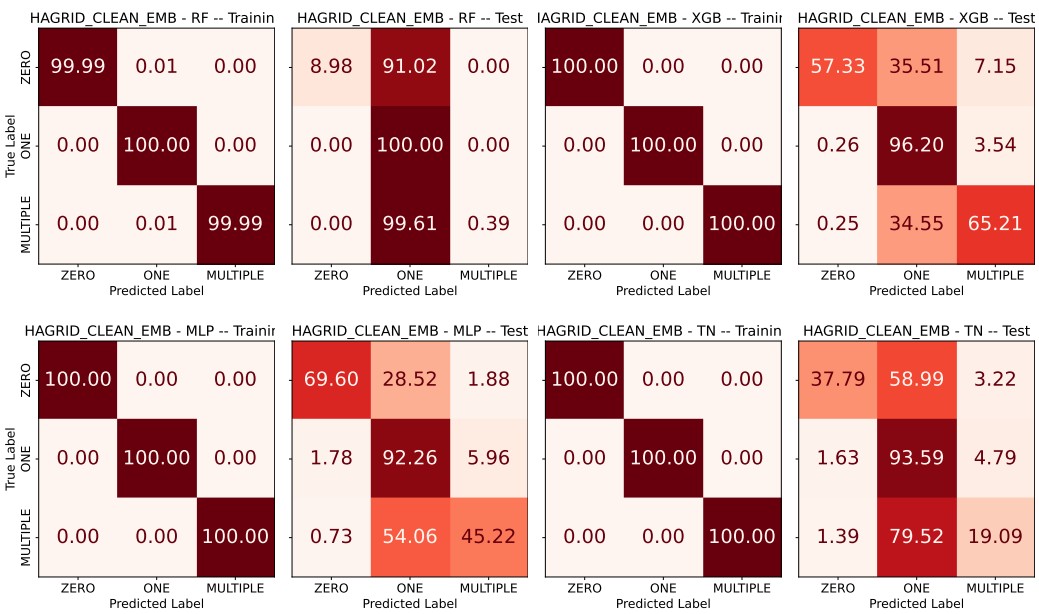

Figure 6: Average confusion matrices obtained in each experiment. The matrices have been normalizes in each row, highlighting the accuracy in each class.

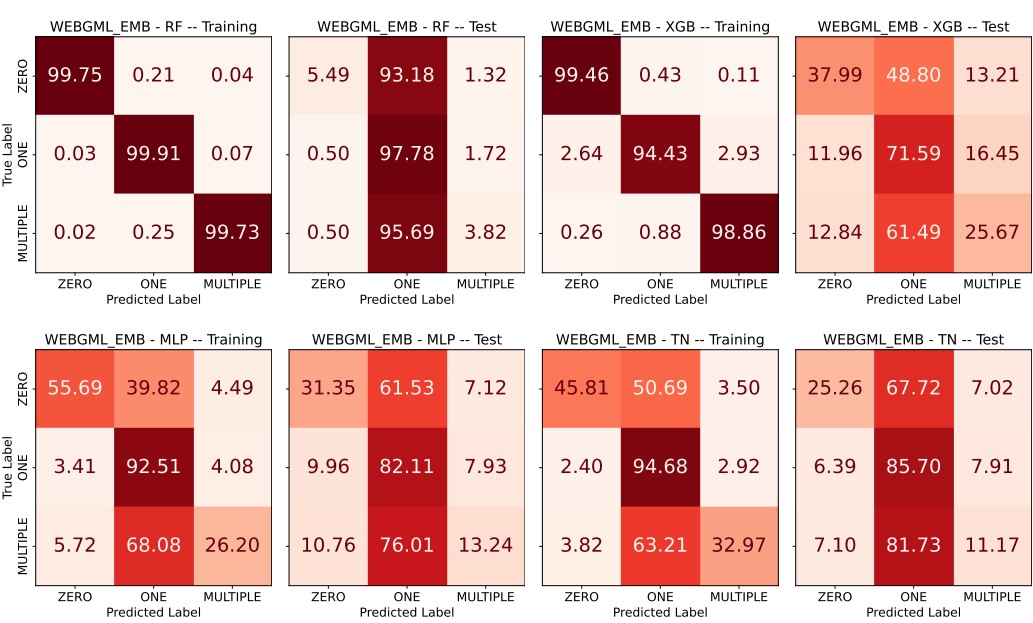

Figure 7: Average confusion matrices obtained in each experiment. The matrices have been normalizes in each row, highlighting the accuracy in each class.

Table 13: Accuracy of each attribution algorithm when using the true label of the pre-attribution (PA) dataset to dictate the desired number of quotes (roofline); top-1 accuracy, and accuracy when using XGB for PA of each algorithm (normalized to the roofline). The XGB results in this table are a 30-run average. Table 13 also includes the HAGRID and WebGLM-QA results. This table further motivates the need for proper attribution datasets since training the pre-attention models in them vastly raises the attribution capacities of the system. In particular, pre-attribution can improve the attribution accuracy of BM25 in the WebGLM-QA dataset to 96.74%, an almost perfect accuracy.

| Attribution Method:
Number of sentences returned: | SC1
1 | SC2 | BM25
1, 2 | MMR | Fuzzy | SPLADE
0, 1, 2 |
|---|---|---|---|---|---|---|
| **HAGRID-Clean** | | | | | | |
| Attribution accuracy (True Label for PA) | | | | | | |
|   Accuracy on the dataset | 76.44 | 77.91 | 71.69 | 75.96 | 59.56 | 78.11 |
| Attribution top-1 accuracy | | | | | | |
|   Normalized | 92.78 | 91.03 | 91.31 | 93.36 | 90.31 | 93.79 |
| Attribution accuracy (XGB for PA) | | | | | | |
|   Training (normalized) | 99.71 | 99.35 | 98.98 | 98.96 | 98.76 | 98.90 |
|   Test (normalized) | 97.87 | 96.98 | 95.95 | 95.67 | 94.81 | 95.88 |
|     Improvement vs top-1 | **+5.09** | **+5.96** | **+4.64** | **+2.30** | **+4.50** | **+2.09** |
| **HAGRID** | | | | | | |
| Attribution accuracy (True Label for PA) | | | | | | |
|   Accuracy on the dataset | 63.62 | 67.06 | 63.72 | 67.69 | 53.30 | 69.19 |
| Attribution top-1 accuracy | | | | | | |
|   Normalized | 85.41 | 81.03 | 78.44 | 80.28 | 76.32 | 80.58 |
| Attribution accuracy (XGB for PA) | | | | | | |
|   Training (normalized) | 99.89 | 99.79 | 99.73 | 99.72 | 99.82 | 99.78 |
|   Test (normalized) | 87.75 | 81.79 | 77.34 | 77.93 | 75.69 | 78.32 |
|     Improvement vs top-1 | **2.34** | **0.76** | **-1.10** | **-2.35** | **-0.63** | **-2.26** |
| **WebGLM-QA** | | | | | | |
| Attribution accuracy (True Label for PA) | | | | | | |
|   Accuracy on the dataset | 71.46 | 78.85 | 96.74 | 79.58 | 62.94 | 89.64 |
| Attribution top-1 accuracy | | | | | | |
|   Normalized | 76.43 | 69.27 | 64.39 | 68.64 | 63.38 | 66.18 |
| Attribution accuracy (XGB for PA) | | | | | | |
|   Training (normalized) | 99.26 | 99.21 | 99.03 | 99.02 | 99.28 | 99.05 |
|   Test (normalized) | 77.00 | 68.52 | 59.62 | 63.30 | 61.84 | 61.16 |
|     Improvement vs top-1 | **0.57** | **-0.75** | **-4.77** | **-5.09** | **-1.54** | **-5.02** |
| Attribution accuracy (XGB for PA) | | | | | | |
|   Trained on HAGRID-Clean (normalized) | 77.05 | 69.91 | 64.07 | 68.23 | 63.05 | 65.76 |
|     Improvement vs top-1 | **0.62** | **0.64** | **-0.32** | **-0.16** | **-0.33** | **-0.42** |

# E    USING SAFE

## E.1    DOWNLOADING AND RUNNING SAFE

The SAFE framework can be downloaded and used using the following steps in a terminal session:

```
% git clone https:[hidden due to double-blind policies]
% cd SAFE/
% python setup_HAGRID_Clean.py
% cd SAFE_FullSystem/
% wget -O documents/filename1.pdf https://arxiv.org/pdf/[your
favorite paper]
% wget -O documents/filename2.pdf https://arxiv.org/pdf/[your
other favorite paper]
% python main.py -i filename1.pdf filename2.pdf -q "Make a summary
of this paper"
```

Be sure to check *SAFE_FullSystem/Conversation_GPT.py* and set up your API Key properly before running *main.py*. In this example, we use the OpenAI API, but this file can be modified to use a different API.

```
import os
from openai import OpenAI
# Set your API key (or set it as an environment variable
OPENAI_API_KEY)
client = OpenAI(api_key=os.getenv("OPENAI_API_KEY"))
```

## E.2    REAL-WORLD APPLICATIONS

Use Case 1: Prompting the system to answer a question about medicine based on the input document, a paracetamol leaflet (search space = 58 sentences). The attribution in the first generated sentence is wrong, but the user's input resulted in a correction. Later sentences were correctly attributed

---

I'm not feeling well, and my doctor said I should take a pill of
paracetamol.  Is it safe to take another pill?
Reply one sentence at a time and wait for me to say ok before showing the
next sentence.
After answering, say DONE

---

ChatGPT Answer:  You should not take more than 2 tablets of paracetamol
(500 – 1000 mg) at a time.
System:  This sentence is attributed to -- "You should take no more
than 8 paracetamol containing tablets or capsules (4000mg or 4g of
paracetamol) in 24 hours ."
USER: Is that in the document?
ChatGPT Answer:  Yes, the document states, "You should not take more than
2 tablets at a time."
System:  This sentence is attributed to -- "You should not take more than
2 tablets at a time."
USER: ok
ChatGPT Answer:  A gap of at least 4 hours should be left between doses.
System:  This sentence is attributed to -- "A gap of at least 4 hour
should be left between doses."
USER: ok
ChatGPT Answer:  The maximum amount you should take in a 24-hour period
is 8 tablets (4000 mg or 4 grams).
System:  This sentence is attributed to -- "A maximum of 8 tablets
(4000mg or 4g) should be taken in a 24 hour period."
USER: ok
ChatGPT Answer:  If you weigh less than 50 kg (around 7 1/2 stone), you
may need a reduced dose and should talk to your doctor, pharmacist, or
pain specialist about this.
System:  This sentence is attributed to -- "If you weigh less than 50kg
(around 7 1/2 stone) then you may need a reduced dose."
USER: ok
ChatGPT Answer:  Paracetamol is generally safe if taken as prescribed,
but taking more than the recommended dose can cause damage to your liver.
System:  This sentence is attributed to -- "It is dangerous to take more
paracetamol than recommended and it can cause damage to your liver."
USER: ok
ChatGPT Answer:  If you have problems with your liver or drink a lot of
alcohol, you should talk to your doctor, pharmacist, or pain specialist
before taking paracetamol.  DONE
System:  This sentence is attributed to -- "If you have problems with
your liver or drink a lot of alcohol you should talk to your doctor,
pharmacist or pain specialist about this before taking paracetamol or
other tablets with paracetamol in them."

---

Use Case 2: Prompting the system to answer a question about the content of a paper (Bang et al., 2025) (search space = 1272 sentences). In this case, the attribution algorithm does not find a quote stating that ambiguous or incomplete input prompts may lead to hallucinations, but we notice that GPT seems to have a wrong idea about the meaning of "extrinsic" in the context of the paper.

---

```
How can we induce hallucinations in LLM outputs?
Reply one sentence at a time and wait for me to say ok before showing the
next sentence.
After answering, say DONE
```

---

ChatGPT Answer:  Hallucinations in LLM outputs can be induced by
providing ambiguous or incomplete input prompts.
System:  This sentence is attributed to -- "Extrinsic hallucinations,
where the generated content is not consistent with the training data, are
increasingly important as LLMs evolve."
USER: Are there examples in the paper?
ChatGPT Answer:  The paper provides examples illustrating hallucinations,
such as generated content contradicting the source input in intrinsic
hallucinations.
System:  This sentence is attributed to -- "Intrinsic hallucinations are
generated texts that contradict the source query."
USER: Ok
ChatGPT Answer:  It also discusses examples related to extrinsic
hallucinations, where generated outputs are inconsistent with the
training data or unverifiable by the input context.
System:  This sentence is attributed to -- "Meanwhile, input sources that
are not self-consistent or that contradict the training data (Ming et al.,
2024; Filippova, 2020) can result in intrinsic hallucinations."
USER: Ok
ChatGPT Answer:  For instance, the paper mentions questions about the
"latest Summer Olympics," where outputs may be factually incorrect but
not hallucinated if consistent with training data.
System:  This sentence is attributed to -- "In some cases, the output may
be factually incorrect but not hallucinated if it remains consistent with
the training data."
USER: Why?
ChatGPT Answer:  Because hallucination is defined as inconsistency with
the model's training data or input context, so an output consistent with
training data, even if factually outdated or incorrect in the real world,
is not considered a hallucination.
System:  This sentence is attributed to -- "In some cases, the output may
be factually incorrect but not hallucinated if it remains consistent with
the training data."
USER: Ok
ChatGPT Answer:  In contrast, hallucinated content may be factually
correct with respect to external sources, but is still considered
hallucination if it deviates from the input context or training data.
DONE
System:  This sentence is attributed to -- "In contrast, hallucinated
content can be factual even if it deviates from user input, as long as it
remains consistent with external reference knowledge sources."
```

---

Use Case 3: Prompting the system to answer a question about the content of a paper (Zhang et al., 2025), using *GPT-4.1-mini* (search space = 765 sentences). Interestingly, we find that attribution not only validates the generated sentences but also explains them from a different perspective.

---

How can we induce hallucinations in LLM outputs?
Reply one sentence at a time and wait for me to say ok before showing the next sentence.
After answering, say DONE

---

ChatGPT Answer:  Hallucinations in LLM outputs can be induced by providing ambiguous or incomplete input prompts.
System:  This sentence is attributed to -- "In the general domain, input-conflicting hallucinations occur when the answers generated by LLMs deviate from the original intentions of user inputs [ 19]."
USER: ok
ChatGPT Answer:  Another way is to introduce conflicting or incorrect contextual information in the prompt to confuse the model.
System:  This sentence is attributed to -- "This hallucination may result from the model's incorrect parsing or inaccurate internal representation of the input information, causing the output content to deviate from the intent and context of the source input."
USER: ok
ChatGPT Answer:  Introducing requirements that rely on knowledge missing from the model's training data can also lead to hallucinations.
System:  This sentence is attributed to -- "We identify four potential factors that cause hallucinations:  training data quality, intention understanding capacity, knowledge acquisition capacity, and repository-level context awareness."
USER: ok
ChatGPT Answer:  Complex contextual dependencies, such as those in repository-level code generation involving user-defined functions or non-code resources, can induce hallucinations due to the model's limited context awareness.
System:  This sentence is attributed to -- "In contrast, we investigate hallucinations in repository-level code generation within more practical development contexts, adopting a holistic perspective that encompasses phenomena ,mechanism , and mitigation ."
USER: ok
ChatGPT Answer:  Additionally, the hallucinations may arise when the model lacks proper intention understanding, leading to task requirement conflicts.
System:  This sentence is attributed to -- "Most hallucinations associated with Task Requirement Conflicts andFactual Knowledge Conflicts can be, to a certain extent, attributed to data quality issues in the training corpora."
USER: ok
ChatGPT Answer:  Lastly, low-quality or insufficient training data related to the specific domain or task can cause the model to generate hallucinated content.
DONE
System:  This sentence is attributed to -- "This hallucination may result from the model's incorrect parsing or inaccurate internal representation of the input information, causing the output content to deviate from the intent and context of the source input."

---

Use Case 4: Prompting the system to explain how to calculate the force of gravity on Earth's surface based on the Wikipedia pages for Gravity and Earth, using *GPT-5-mini* (search space = 3458 sentences). In this case, SAFE produces satisfactory results, but we notice that the method used to extract sentences from PDFs cuts sentences in unexpected places.

---

Based on the input documents, show me step by step how to calculate the force of gravity on Earth.  All values and equations should be in plain text.  Reply one sentence at a time and wait for me to say ok before showing the next sentence.
After answering, say DONE

---

ChatGPT Answer:  We will calculate the gravitational force on an object at Earth's surface using Newton's law of universal gravitation and the Earth parameters given in the document.
System:  This sentence is attributed to -- "The force of gravity experienced by objects on Earth's surface is the vector sum of two forces:[6] (a) The gravitational attraction in accordance with Newton's universal law of gravitation, and (b) the centrifugal force, which results from the choice of an earthbound, rotating frame of reference."
USER: ok
ChatGPT Answer:  Start with Newton's law of universal gravitation:  F = G * M * m / r$^2$.
System:  This sentence is attributed to -- "[44][45] Newton's formulation was later condensed into the inverse-square law:  where F is the force, m1 and m2 are the masses of the objects interacting, r is the distance between the centers of the masses and G is the gravitational constant 6.674 * 10-11 m3 kg-1 s-2."
USER: ok
ChatGPT Answer:  Use values:  G = 6.67430e-11 m$^3$ kg$^{-1}$ s$^{-2}$, M = 5.972168e24 kg (Earth mass from the document), r = 6,371,000 m (mean radius converted from 6,371.0 km), and let m be the object's mass in kg.
System:  This sentence is attributed to -- "Among the Solar System's planetary-sized objects, Earth is the object with the highest density. Earth's mass is approximately 5.97 × 1024 kg (5.970 Yg)."
USER: ok
ChatGPT Answer:  Calculate the gravitational acceleration at Earth's surface:  g = (6.67430e-11 m$^3$ kg$^{-1}$ s$^{-2}$ * 5.972168e24 kg) / (6,371,000 m)$^2$ ≈ 9.82 m/s$^2$.
System:  No attribution needed.
USER: ok
ChatGPT Answer:  For any mass m (in kilograms), the gravitational force at Earth's surface is F = m * g, so numerically F (in newtons) ≈ 9.82 * m.
System:  This sentence is attributed to -- "Near Earth's surface, gravitational acceleration is approximately 9.8 m/s2 (32 ft/s2)."
USER: ok
ChatGPT Answer:  For example, for m = 1 kg, the gravitational force F ≈ 9.82 N.
System:  This sentence is attributed to -- "The force of gravity, F is written using the gravitational constant, G, as[7] for two masses, m, and m' separated by a distance r.  Gravity is considered to be one of four fundamental interactions."
USER: ok, thank you
ChatGPT Answer:  You're welcome -- DONE

---

