# OpenReview forum: "SAFE: Improving LLM Systems using Sentence-Level In-generation Attribution"
_ICLR.cc/2026/Conference — ICLR 2026 Conference Withdrawn Submission_

### Official Review · Reviewer_EhhC · 2025-10-18

**Soundness:** 2
**Presentation:** 3
**Contribution:** 2
**Rating:** 2
**Confidence:** 3

**Summary:**

This paper introduces SAFE, a sentence-level in-generation attribution framework for RAG systems. The framework is model-agnostic and can be deployed on any downstream task of any LLM. For each sentence generated by the LLM, it searches for appropriate references to help users verify the reliability of the generated content. The framework consists of two main steps: when the LLM generates a sentence, a small classifier predicts how many citations (0/1/>1) are needed, and then a standardized attributor assigns the sentence to the most relevant reference sentences.

**Strengths:**

* The paper presents a clear and practical engineering work.
* The provided code repo is easy to read and use, showing good reproducibility.
* The framework can be effectively applied on low-end machines, such as personal devices that only have API access to LLMs.
* The authors also contribute a manually cleaned version of the HAGRID dataset.

**Weaknesses:**

* **The proposed framework lacks novelty.** It mainly combines two steps, pre-attribution (predicting the number of citations) and attribution (assigning references). This limits the paper’s contribution to an engineering solution instead of a innovation.
* **The technical methods used are quite standard and outdated.** For pre-attribution, the authors test Random Forest (RF), XGBoost (XGB), Multi-Layer Perceptron (MLP), and TabularNet (TN), all of which are common lightweight models. The attribution part also relies on conventional techniques.
* **The experimental section is not convincing.** There is a lack of comparison with existing solutions, and the real-world application section only provides example-level demonstrations, which do not sufficiently prove the framework’s effectiveness.

**Questions:**

As noted in the limitation section, sentence-level attribution has an inherent issue: retrieved sentences often lose contextual meaning, which may cause misinterpretation in certain cases. Conversely, a single generated sentence by the LLM can be very complex and may require multiple references. I hope the authors can further explain how these issues are mitigated in downstream tasks, especially when using the proposed framework.

---

### Official Review · Reviewer_mk1a · 2025-10-30

**Soundness:** 1
**Presentation:** 3
**Contribution:** 1
**Rating:** 0
**Confidence:** 4

**Summary:**

The paper introduces SAFE, a Sentence-level Attribution Framework for RAG systems designed to improve the trustworthiness and verifiability of LLM outputs. SAFE attributes generated sentences using a light-weight design: First is through a Pre-attribution step: A classifier predicts whether a generated sentence requires zero, one, or multiple references; Then a "Attribution" step: Based on the pre-attribution prediction, the system selects an appropriate attribution algorithm (such as embedding-based methods, fuzzy string matching, BM25, or SPLADE) to find the matching quote(s) from the source document.

**Strengths:**

1. Open source framework that is very light weight
2. HAGRID-Clean Dataset: The authors provide a manually cleaned version of the HAGRID attribution dataset to address issues with noisy data and inconsistent referencing in existing benchmarks. (not sure this dataset will be released, though)

**Weaknesses:**

1. Lack comparison to various methods working on grounded generation, for example directly use NLI model [1], or more advanced methods([2], [3], just to named a few.)
2. The evaluation and comparison is on the same distribution, showing little generalization testing
3. The real-world testing section do not have evaluation results


[1] Tianyu Gao, Howard Yen, Jiatong Yu, and Danqi Chen. Enabling large language models to generatetext with citations. In Proceedings of the 2023 Conference on Empirical Methods in Natural Language Processing.
[2] Hsu, I., Wang, Z., Le, L. T., Miculicich, L., Peng, N., Lee, C. Y., & Pfister, T. (2024). Calm: Contrasting large and small language models to verify grounded generation. arXiv preprint arXiv:2406.05365.
[3] Maheshwari, H., Tenneti, S., & Nakkiran, A. (2025). CiteFix: Enhancing RAG Accuracy Through Post-Processing Citation Correction. arXiv preprint arXiv:2504.15629.

**Questions:**

1. During cleaning the dataset, how do you make sure the consistency on "how many references are needed"? How aligned it is across different annotators?

---

### Official Review · Reviewer_mrMd · 2025-10-31

**Soundness:** 2
**Presentation:** 3
**Contribution:** 2
**Rating:** 2
**Confidence:** 4

**Summary:**

The paper proposes an attribution method for Retrieval-Augmented Generation (RAG) systems. Current RAG systems often provide no citations or cite entire documents, making fact-checking impractical, and the  SAFE framework, does so by having a classifier to determine the number of references it requires and then uses different attribution methods for references.

To train the pre-attribution classifier, the authors created HAGRID-Clean, a manually cleaned and re-labeled version of the HAGRID dataset, which they identified as being too noisy for this task. Their results show the PA classifier achieves ~95% accuracy on this new dataset and that the full SAFE framework improves the final attribution accuracy by 2.1–6.0% (normalized) compared to a standard top-1 baseline.

**Strengths:**

1. **Well written paper:** The paper is very thorough and easy to follow and reproduce if wanted.
2. **Data Contribution:** The creation of the HAGRID-Clean dataset is a good resource for the community. The authors identified that existing attribution datasets suffer from noise, such as "over-referencing" and "under-referencing". By manually reviewing and re-labeling the data based on the ideal number of references, they created a high-quality benchmark.
3. **Strong Empirical Validation:** The framework consistently improves the (normalized) accuracy of all tested attribution algorithms (e.g., +5.96% for SC2, +2.09% for SPLADE) over a simple top-1 baseline.

**Weaknesses:**

1. **Misleading "In-Generation" Claim and Outdated Baselines:** The paper's central claim of "in-generation" attribution is misleading. The framework operates post-hoc on a per-sentence basis; it classifies and attributes a fully generated sentence. This is fundamentally different from true in-generation attribution models (e.g., Gao et al., 2023; RARR) that co-generate text and citation markers. By not comparing against this dominant and more recent line of research (listed a few below), the paper's evaluation is disconnected from the current state-of-the-art. The baselines used (e.g., Fuzzy String Matching, BM25) are classical IR methods and do not represent strong, modern attribution baselines.

2.  **High Risk of False Positives (Poor "No Quote Found" Detection):** The framework's reliability is critically undermined by its inability to handle ungrounded sentences. The authors note that most of their chosen attributors (SC1, SC2, BM25, MMR) *always* return a quote, even if it's irrelevant. This forces the system to rely almost entirely on the pre-attribution classifier's "ZERO" class to filter un-attributable content. If the PA classifier makes a mistake (e.g., classifies a hallucination as "ONE"), the system will *always* find and present an incorrect attribution, creating a false positive and violating the system's core goal of building trust.

3.  **Fragile Pre-Attribution Classifier:** The best-performing PA classifier relies on a set of 24 hand-crafted textual features. This approach is notoriously fragile and prone to poor generalization. The fact that sentence embeddings performed significantly worse on the HAGRID-Clean dataset is highly concerning. It suggests the classifier is not learning the *semantic complexity* of a sentence but rather overfitting to superficial, stylistic heuristics present in the HAGRID-Clean data. This classifier is unlikely to be robust when applied to text from different domains (e.g., legal, medical) or output from more advanced LLMs.

4.  **Limited Benchmark Evaluation:** The entire evaluation is conducted on HAGRID and WebGLM-QA. While the creation of HAGRID-Clean is a good contribution, the paper fails to evaluate its framework on other standard attribution or open-domain QA benchmarks (e.g., ASQA). This makes it impossible to contextualize its performance against the wider field of attribution research and assess its generalizability.

5.  **Impractical Interaction Model and Unsubstantiated Latency Claims:** The proposed user-in-the-loop system requires a synchronous, blocking confirmation from the user after *every single sentence*. This would be impractically slow and tedious for any real-world application. The paper provides no discussion of the significant latency this introduces or any user studies to validate that this interaction is genuinely preferred. While the framework is designed for "low-end machines, the claims about latency are unsubstantiated.

[1] RARR: Researching and Revising What Language Models Say, Using Language Models. Gao et al., 2022

[2] Enabling Large Language Models to Generate Text with Citations. Gao et al., 2023.

[3] Attribute First, then Generate: Locally-attributable Grounded Text Generation. Slobodkin et al., 2024

[4] GenerationPrograms: Fine-grained Attribution with Executable Programs. Wan et al., 2024

[5] ContextCite: Attributing Model Generation to Context, Cohen-Wang et. al., 2024

**Questions:**

1.  **Comparison to SOTA:** How does SAFE's performance (in both attribution accuracy and end-to-end latency) compare to generative attribution models (e.g., Gao et al., 2023; RARR) that generate citations directly as part of the text? Why were these state-of-the-art methods, which are also "in-generation," omitted as baselines?

2.  **Dataset Annotation Quality:** Regarding the HAGRID-Clean dataset, what was the detailed annotation protocol? What were the qualifications of the human reviewers, and what was the inter-annotator agreement (IAA) for assigning the "ideal" number of references? This is critical for assessing the validity of the new dataset.

3. **Robustness of Pre-Attribution:** Given that the textual-feature-based classifier outperformed embeddings on HAGRID-Clean, what evidence suggests it will generalize? Have you tested its robustness on out-of-domain data or text from more capable LLMs (e.g., GPT-4), which may not share the same stylistic artifacts as the dataset?

4.  **Usability and User Preference:** Was any user study conducted to validate the sentence-by-sentence confirmation loop? Is this synchronous interaction truly preferred by users over a fully generated, asynchronously citable response? Furthermore, does the interactive correction shown in Use Case 1 measurably improve the factual groundedness of the LLM's *subsequent* sentences?

---

### Official Review · Reviewer_hF1X · 2025-11-01

**Soundness:** 2
**Presentation:** 2
**Contribution:** 3
**Rating:** 2
**Confidence:** 3

**Summary:**

This paper addresses the need for user-verifiable outputs in Retrieval-Augmented Generation (RAG) systems. The authors argue that providing entire documents is impractical for users who need to fact-check an LLM's response. Therefore, the paper proposes SAFE, a Sentence-level Attribution Framework that operates in-generation, providing users with the most relevant quotes from source documents as each sentence is being generated. This allows for immediate verification and correction.

A secondary contribution is the manual cleaning and re-labeling of the HAGRID dataset to create HAGRID-Clean. This new dataset addresses noise and inconsistent labeling in the original, providing a more reliable benchmark for sentence-level attribution tasks.

The SAFE pipeline itself consists of two main stages: (1) train a "Pre-attribution Classifier" (PA) that predicts the number of quotes required (0, 1, or 2+) for a given sentence, and (2) select and operate a set of attribution algorithms (e.g., SPLADE, embedding similarity) that retrieve the actual quotes.

**Strengths:**

The proposal of an "in-generation" (sentence-by-sentence) attribution system is a novel and highly practical contribution. This design directly addresses the user-experience bottleneck of post-hoc, document-level verification and empowers users to correct hallucinations in real-time.

The use of a lightweight Pre-attribution (PA) classifier first predicts the required number of quotes is an efficient design. As shown in Table 5, this two-step process improves the accuracy of the final attribution by 2.1-6.0% (normalized) over a simple top-1 retrieval baseline, optimizing the pipeline for its real-time goal.

Finally, the authors creates HAGRID-Clean by manually cleaning, merging, and re-labeling the original HAGRID dataset, to provide a valuable, high-quality resource for future research in this specific domain.

**Weaknesses:**

The two-step design creates its own limitations. As the paper notes, the system's ability to identify sentences requiring zero attribution is weak, as most attributors are not optimized for "no match" detection. The system is thus forced to conservatively find a quote, even for common knowledge.

The framework requires a dedicated dataset (like HAGRID-Clean) with sentence-level labels (0, 1, 2+) to train the Pre-attribution Classifier. This multi-stage pipeline, which requires pre-training a classifier on a highly specific dataset, presents a significant barrier to adoption compared to simpler, training-free retrieval methods.

A major concern is the "roofline" accuracy in Table 5. Even when the Pre-attribution Classifier is perfect (using the "True Label"), the best-performing attributor (SPLADE) only achieves 78.11% accuracy. This suggests the core retrieval algorithms are a significant bottleneck, and even a perfect classifier cannot compensate for their low performance.

The system is strictly sentence-level, which may not attribute generated sentences that are synthesized from multiple, non-contiguous source sentences or that rely on the surrounding context of a quote for their meaning. The paper's own limitations section acknowledges this.

**Questions:**

* The evaluation metric (Section 3.2.3) is very strict, requiring a match in the number of quotes (0, 1, or 2+) and a subset match of the quotes. Would a standard IR metric like Recall@k be a more practical measure?

* The "roofline" accuracy in Table 5 at ~78% seems low, implying the core attributors are the main bottleneck. Given this low ceiling, what do the authors see as the most promising path forward: developing better attributors?

* Could the authors comment on the latency vs. accuracy trade-off of using a more powerful, small LLM (e.g., 3B-8B parameters) as PA+the attributor themselves, rather than the proposed two-step process?

* The limitations section notes that sentence-level attribution can miss context. Has a hybrid approach been considered, such as retrieving a "chunk" (e.g., the surrounding +/- n sentences) to present to the user alongside the primary attributed quote?

---

### Note · Authors · 2025-11-13

I have read and agree with the venue's withdrawal policy on behalf of myself and my co-authors.